# Broadband angular spectrum differentiation using dielectric metasurfaces

Ming Deng[1,6], Michele Cotrufo [2,3,6], Jian Wang[1], Jianji Dong[1], Zhichao Ruan [4], Andrea Alù [2] ✉ & Lin Chen [1,5] ✉

Signal processing is of critical importance for various science and technology fields. Analog optical processing can provide an effective solution to perform large-scale and real-time data processing, superior to its digital counterparts, which have the disadvantages of low operation speed and large energy consumption. As an important branch of modern optics, Fourier optics exhibits great potential for analog optical image processing, for instance for edge detection. While these operations have been commonly explored to manipulate the spatial content of an image, mathematical operations that act directly over the angular spectrum of an image have not been pursued. Here, we demonstrate manipulation of the angular spectrum of an image, and in particular its differentiation, using dielectric metasurfaces operating across the whole visible spectrum. We experimentally show that this technique can be used to enhance desired portions of the angular spectrum of an image. Our approach can be extended to develop more general angular spectrum analog meta-processors, and may open opportunities for optical analog data processing and biological imaging.

Signal processing is vital for several science and engineering disciplines. The increasing demand for high-speed and high-efficiency signal processing has inspired scientists to develop faster, integrated and efficient devices that can process signals and images[1–3]. Digital signal processing based on integrated electric circuits can perform complex data processing, but it needs an analog-to-digital converter to discretize the input analog signal into a series of bits. After being processed according to the desired operation, the discrete signals are converted back into analog signals through a digital-to-analog converter. This method suffers from low-speed, large memory requirements and high-power consumption, originating from complicated conversion process[4] and hardware bottleneck[5,6].

Optical analog processing, with the ability to perform massive parallel processing with high-speed and low-power consumption, has gained significant attention[7–9]. Optical analog processing has been demonstrated both in the spatial and temporal domain[10–14] with, e.g., conventional 4 F systems, interferometers and Soleil Babinet compensators. The first-order temporal differentiator was realized by introducing different time delays for two mutually orthogonally polarized components through a Soleil Babinet compensator[14]. However, the resultant optical systems are bulky, and hence are against high-density optical integration. Conventional optical elements do not offer the opportunity to spatially and continuously vary transmission amplitude and phase at the sub-wavelength scale. Metasurfaces, planar arrays composed of subwavelength artificial meta-atoms, provide

[1]Wuhan National Laboratory for Optoelectronics and School of Optical and Electronic Information, Huazhong University of Science and Technology, Wuhan 430074, China. [2]Photonics Initiative, Advanced Science Research Center, City University of New York, New York, NY 10031, USA. [3]The Institute of Optics, University of Rochester, Rochester, NY 14627, USA. [4]School of Physics, Zhejiang Province Key Laboratory of Quantum Technology and Device, and State Key Laboratory for Extreme Photonics and Instrumentation, Zhejiang University, Hangzhou 310027, China. [5]Shenzhen Huazhong University of Science and Technology Research Institute, Shenzhen 518063, China. [6]These authors contributed equally: Ming Deng, Michele Cotrufo. ✉e-mail: aalu@gc.cuny.edu; chen.lin@mail.hust.edu.cn

opportunities to manipulate light by locally tailoring the amplitude, phase and polarization with much more flexibility. A number of compact optical devices have been developed using metasurfaces, such as lenses[15–18], holography systems[19–22], polarimeters[23,24], polarization elements[25,26], retroreflectors[27–29] and optical imaging encoding[30,31]. Recently, the research interest has shifted towards practical applications of metasurfaces, such as high-quality imaging[32–35] and optical analog computation[5,8,9,36–46]. A wide range of optical analog computations using metasurfaces, such as spatial imaging differentiation[5,8,37–39,43–45], integration[37,41,47], convolution[37,42] and equation solving[36,40], have been successfully demonstrated. Spatial differentiation has also been realized using photonic crystals[48,49], spin Hall effect[50,51], and surface plasmon-based devices[52]. Realizing spatial differentiation requires a transfer function featuring a zero at normal incidence, in order to suppress the average of the input image and enhance its edges. Generally, most strategies rely on resonant mechanisms to generate this transmission dip. As a result, they often work only in a narrow band of frequencies. Recent works have proposed ways to overcome this challenge based on dispersion engineering[46]. Bragg gratings and microring resonators can also provide a compact method to enable temporal differentiation[53,54]. The working bandwidth of microring resonators however is intrinsically low, smaller than those based on Bragg gratings and interferometers[53,54]. The limited operation bandwidth may restrict the capability of exploiting wavelength division multiplexing to enhance data processing speed, which is important for large-scale and high-speed optical analog computation[55,56]. Spatial and temporal differentiation have been exploited for applications in edge detection, optical processing of microwave signals, analog-digital conversion, pulse shaping, and dark-soliton detection[5,8,12,13,38,39,43,44,50–53].

Fourier optics provides a powerful approach to optical processing, by filtering the frequency components of the angular spectrum of the input image. In a conventional 4 F system, the image to be processed is decomposed into its spatial Fourier components by a first optical lens. Various components to filter the spatial frequencies, made by bulky diffractive optical elements, including pinholes, opaque dots, masks and phase plate shifters[57], can then be exploited for angular spectrum filtering of the incoming waves, leading to imaging applications, such as dark-field[58,59], schlieren[60,61], and phase-contrast imaging[62]. However, imparting mathematical operations on the angular spectrum has not been explored to the best of our knowledge. In this paper, we demonstrate metasurfaces that perform mathematical operations on the angular spectrum of an image - "angular spectrum analog meta-processors"—that can be used to perform optical analog processing of the Fourier spectrum—such as differentiation, integration, convolution, and other linear operations as light propagates through them. Depending on the optical operations of interest for angular spectrum processing, we retrieve the required amplitude and phase modulation on the incoming light. We demonstrate differentiation of the angular spectrum based on dielectric metasurfaces across the whole visible frequencies, and we apply it to demonstrate novel applications in the angular spectrum domain, i.e., enhancement of desired angular spectrum features. It should be noted that the angular spectrum differentiation could be also realized by using a lens to extract the angular spectrum, followed by a spatial differentiator to complete angular spectrum differentiation. However, this approach would result in a bulky device. In contrast, here we propose a compact approach to realize angular spectrum differentiation by using a single metasurface. The application potential of enhancing desired portions of the angular spectrum of an image has not been previously exploited in the field of optical analog processing.

## Results
### Design principles and simulation results
The angular spectrum $A_p(k_x, k_y)$ of an incoming wavefront can be retrieved by performing the Fourier transform of the in-plane electric field distribution $E_p(x, y)$

$$A_p(k_x,k_y) = F[E_p(x,y)] = \iint_{-\infty}^{+\infty} E_p(x,y) \exp(-ik_x x - ik_y y)dxdy \quad (1)$$

where i is the imaginary unit, $F$ indicates the Fourier transform, $p = x (y)$ labels the x- (y-) component, and $k_x$, $k_y$ are 2D Fourier domain variables along x and y directions, respectively. $A_p(k_x, k_y)$ and $E_p(x, y)$ represent the complex amplitudes of the electric field in the Fourier and spatial domain, respectively. After passing through an optical diffractive element, the angular spectrum is transformed to $\hat{H}A_p(k_x, k_y)$, where $\hat{H}$ is the angular spectrum operation exerted by the optical diffractive element, modulating the amplitude and phase of $A_p(k_x, k_y)$ of the incoming wavefront. If an arbitrary order differentiation $\hat{H} = \sum_{j=1}^{l}[C_j \frac{\partial^{m_j + n_j}}{(\partial k_x)^{m_j}(\partial k_y)^{n_j}}]$ is exerted on the Fourier transform $A_p(k_x, k_y)$, we have

$$\sum_{j=1}^{l}\left[ C_j \frac{\partial^{m_j + n_j}}{(\partial k_x)^{m_j}(\partial k_y)^{n_j}}\right] A_p(k_x,k_y)$$
$$= \iint_{-\infty}^{+\infty} \sum_{j=1}^{l}[C_j(-ix)^{m_j}(-iy)^{n_j}]E_p(x,y) \exp(-ik_x x - ik_y y)dxdy \quad (2)$$

where $j$ indicates the differentiation type, and $l$ denotes the maximum number of differentiation type ($l$ is a positive integer, and $j \in [1, l]$). $m_j$ and $n_j$ are natural numbers labeling the order of the partial derivative with respect to $k_x$ and $k_y$, respectively, and $C_j$ is a complex coefficient (associated with the $j$-th differentiation type), affecting the efficiency and phase delay for the output angular spectrum. Such differential operation in the Fourier domain can be realized by a real-space transfer function $t(x, y) = \sum_{j=1}^{l}\left[ C_j(-ix)^{m_j}(-iy)^{n_j}\right]$, which is related to $\hat{H}$ via Eq. (2). In other words, we assume that the input image, described by the spatial domain electric field $E_p(x, y)$, is filtered by a spatially varying mask with transmission amplitude and phase. We first discuss the general properties of the required mask, and then we show how this real-space transfer function can be achieved with a metasurface. For example, if $l = 1$, there is only one term in Eq. (2), associated with $(m_j, n_j) = (m_1, n_1)$ and $C_1$. If $(m_1, n_1)$ is chosen as (1, 1), then $\hat{H}$ is equal to $C_1 \frac{\partial^2}{\partial k_x \partial k_y}$, associated with $t(x, y) = -C_1 xy$. The input angular spectrum $\hat{H}$ will contain second-order partial derivatives with respect to $k_x$ and $k_y$, respectively. If $(m_1, n_1) = (1, 0)$, $\hat{H} = C_1 \frac{\partial}{\partial k_x}$ with $t(x, y) = -iC_1 x$, the input angular spectrum will merely consist of the first-order partial derivative with respect to $k_x$. If $l = 2$, there are two types of differentiation, associated with $(m_j, n_j) = (m_1, n_1)$ and $(m_2, n_2)$, and $C_1$, $C_2$. By choosing $(m_1, n_1) = (1, 0)$, $(m_2, n_2) = (0, 1)$ and $C_1 = C_2$, $\hat{H} = C_1 \frac{\partial}{\partial k_x} + C_1 \frac{\partial}{\partial k_y}$ with $t(x, y) = -iC_1(x + y)$, a sum over the two first-order partial derivatives with respect to $k_x$ and $k_y$, will be applied on the input angular spectrum. The values of $l$, $C_j$ and $(m_j, n_j)$ used for the three types of differentiation can be found in Supplementary Table S1 of Supplementary Note 1. The real-space transfer function $t(x, y)$ provided by the optical diffractive elements should be proportional to $xy$, $x$ and $x + y$, respectively, to realize these three types of differentiation. It should be emphasized that, in principle, we can also realize linear combinations of higher-order partial derivatives by flexibly choosing $l$, $C_j$, and $(m_j, n_j)$. For example, we can achieve $\hat{H} = C_1 \frac{\partial^3}{\partial k_x^3} + C_1 \frac{\partial^3}{\partial k_y^3}$ with $t(x, y) = iC_1(x^3 + y^3)$, associated with $l = 2$, $(m_1, n_1) = (3, 0)$, $(m_2, n_2) = (0, 3)$ and $C_1 = C_2$, and $\hat{H} = C_1 \frac{\partial^3}{\partial k_x^3} - iC_1 \frac{\partial^2}{\partial k_y^2} + 2C_1 \frac{\partial}{\partial k_x}$ with $t(x, y) = iC_1(x^3 + y^2 - 2x)$, associated with $l = 3$, $(m_1, n_1) = (3, 0)$, $(m_2, n_2) = (0, 2)$, $(m_3, n_3) = (1, 0)$ and $C_1 = iC_2 = 0.5C_3$.

In the following, we demonstrate metasurfaces tailored to realize these operations. Figure 1a–c schematically show the analog meta-

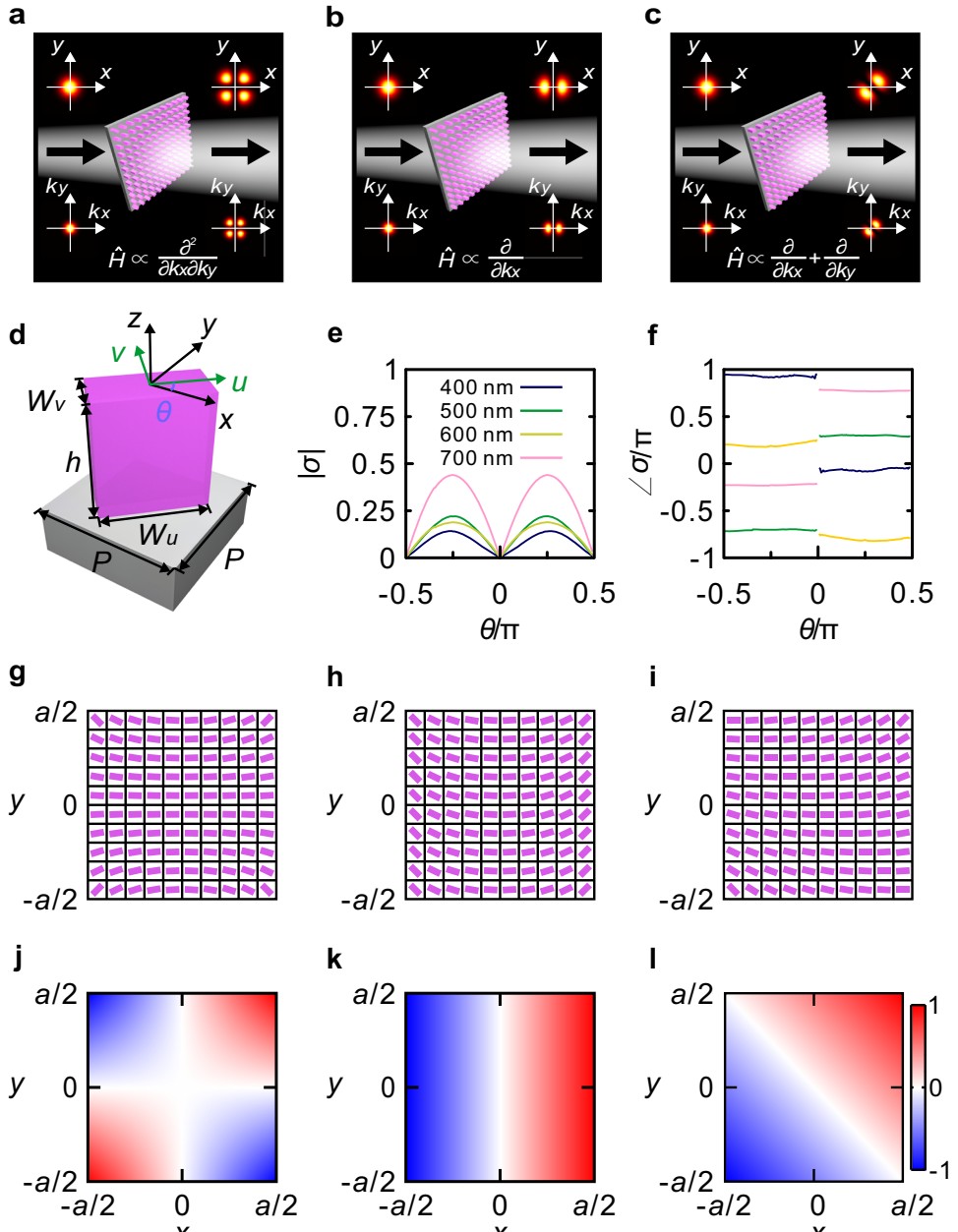

**Fig. 1 | Angular spectrum meta-differentiator. a**–**c** Angular spectrum analog meta-processors for optical analog differentiation processing in the angular spectrum domain, as light propagates through them. The corresponding angular spectrum operation, $\hat{H}$, and complex real-space transfer function, $t(x, y)$, provided by the silicon metasurfaces are, respectively, $\propto \frac{\partial^2}{\partial k_x \partial k_y}$ and $\propto xy$ (**a**), $\propto \frac{\partial}{\partial k_x}$ and $\propto x$ (**b**), and $\propto \frac{\partial}{\partial k_x} + \frac{\partial}{\partial k_y}$ and $\propto x + y$ (**c**). It is assumed that $\hat{C_1}$ is equal to $C_2$ in (**c**). **d** Schematic of the rotated silicon nanopillar on a silica substrate with $W_u = 200$ nm, $W_v = 100$ nm, $h = 220$ nm, and $P = 280$ nm. **e, f** Amplitude (**e**) and phase (**f**) of $\sigma$ versus orientation angle $\theta$. The refractive index and extinction coefficient of silicon were measured by an ellipsometer (see Supplementary Fig. S1 of Supplementary Note 2), and the refractive index of silica is extracted from Ref. 66. **g**–**l** Schematics of the top view presenting the distribution of the orientation angles (**g**–**i**) and distributions of the real parts of the required $\sigma(x, y)/|\sigma(x, y)|_{\max}$ of the three meta-differentiators (**j**–**l**) with $\frac{\partial^2}{\partial k_x \partial k_y}$ (**g, j**), $\frac{\partial}{\partial k_x}$ (**h, k**) and $\frac{\partial}{\partial k_x} + \frac{\partial}{\partial k_y}$ (**i, l**), respectively.

processors that perform the three types of differentiation in the angular spectrum domain. We denote the two opposing edges of the silicon nanopillar cross-section as $W_u$, $W_v$ ($W_u \geq W_v$), namely the long and short sides along $u$ and $v$ directions, respectively. The orientation angle $\theta$ represents the angle between the $x$ axis of laboratory coordinate and $u$ axis (Fig. 1d). The local transmission coefficient for $y$-polarized transmitted light under $x$-polarized incidence is[63] (see Supplementary Note 2 for the derivation)

$$\sigma(x, y) = \frac{(t_u - t_v)}{2} \sin 2\theta(x, y) \qquad (3)$$

where $t_u$ and $t_v$ are the complex transmission coefficients along $u$ and $v$ directions, respectively. Thus, to implement any of the operations discussed above, $\sigma(x, y)$ needs to match the corresponding real-space transfer function $t(x, y)$. The cross sections of the basic building blocks of each nanopillar can be tailored to control $t_u$ and $t_v$ in Eq. (3), regardless of the operating wavelength. In general, $t_u$ and $t_v$ are wavelength-dependent, due to the material dispersion and resonant effects within the metasurface unit cells. In general, $t_u$ and $t_v$ could be chosen to be also position-dependent, in order to increase the flexibility of $\sigma(x, y)$. Here, to keep the design simple, we have chosen to employ silicon nanopillars with a fixed cross section, so that $t_u$ and $t_v$

are wavelength-dependent only. Remarkably, we show that modulating only $\theta$ is sufficient to achieve $\sigma(x, y) = t(x, y)$ for all angular spectrum differential operations considered. Since the required $t(x, y)$ is nonzero at almost every spatial position, the cross section of the silicon nanopillar is chosen to be rectangular, i.e., $W_u > W_v$. For a meta-differentiator composed of a square metasurface with a side length $a$ ($x \in [-a/2, a/2]$, $y \in [-a/2, a/2]$), we show how to define $\theta$ to achieve the aforementioned three types of differentiation. For the first type $t(x, y) = -C_1 xy$, $\theta$ should satisfy $\sin 2\theta = -\frac{2C_1}{t_u - t_v} xy$ with $x \in [-a/2, a/2]$ and $y \in [-a/2, a/2]$. $|C_1| \leq \left| -\frac{2(t_u - t_v)}{a^2} \right|$ holds with $(x, y) = (\pm a/2, \pm a/2)$ since $|\sin 2\theta| \leq 1$. It can be inferred that $C_1$ should be chosen as $\pm \frac{2(t_u - t_v)}{a^2}$ for the purpose of maximizing the efficiency of the output angular spectrum, corresponding to $\theta = \pm \arcsin(4xy/a^2) / 2$. Using a similar method as for the first meta-differentiator, we can also retrieve $\theta = \pm \arcsin(2x / a) / 2$ and $\theta = \pm \arcsin[(x+y) / a] / 2$ for the other two types of meta-differentiators, associated with $t(x, y) = -iC_1 x$ and $t(x, y) = -iC_1 (x+y)$, respectively. The detailed derivation process can be found in Supplementary Note 2. It should be noted that, although the three types of differentiator can work over a wide spectral band, the amplitude and phase shift of the transmissive angular spectrum are wavelength-dependent, as $C_1$ is dependent on the light wavelength. It should also be noted that the actual meta-differentiator has a limited size, and thus it can only process a certain range of angular spectrum. Considering an input beam with a Gaussian profile, the waist radius $w$ allowed has an upper limit, which depends on the size of the meta-differentiator. The detailed analysis regarding the image resolution in the Fourier domain can be found in Supplementary Note 3.

The amplitude of the transmissive angular spectrum is proportional to the amplitude difference between $t_u$ and $t_v$. Here, the cross-sectional geometry of the silicon nanopillar is optimized to obtain a large amplitude difference between $t_u$ and $t_v$ at 450, 532 and 685.5 nm. The silicon layer forming the metasurfaces is deposited by magnetron sputtering in the following experiment, and its refractive index and extinction coefficient is measured by an ellipsometer. The optimization of cross-sectional geometry can be found in Supplementary Fig. S1 of Supplementary Note 2. The simulated amplitude and phase of $\sigma(x, y)$ versus $\theta$ at four wavelengths are shown in Fig. 1e, f, based on finite difference time

domain (FDTD) with the commercial software Lumerical FDTD Solutions. It can be seen that the amplitude of $\sigma(x, y)$ can be changed from zero to a maximum value as $\theta$ is varied from $-0.5\pi$ to $0.5\pi$, and the phase suffers from a $\pi$ shift as $\theta$ changes from a negative value to a positive value. We are able to achieve the aforementioned three types of differentiation by arranging $\theta$ so as to fulfill the required distributions of $\sigma(x, y)$ in a square metasurface. By setting $\theta$ equal to $\arcsin(4xy / a^2) / 2$, $\arcsin(2x / a) / 2$, and $\arcsin[(x+y) / a] / 2$, respectively, the metasurface distributions and the real parts of $\sigma(x, y)/|\sigma(x, y)|_{\max}$ required for the three meta-differentiators are shown in Fig. 1g-i, and Fig. 1j-l, respectively.

To verify the angular spectrum differentiation, we use an $x$-polarized Gaussian beam as the input source to illuminate the meta-differentiators (Fig. 2a), where the associated angular spectrum is shown in Fig. 2b. The corresponding angular spectrum of the filtered beam, given by $|\hat{H} A_x|^2 / |\hat{H} A_x|^2_{\max}$, is theoretically calculated for three types of angular spectrum differentiation (Fig. 2c). Three meta-differentiators with 16.8 μm × 16.8 μm in size (e.g., 60 × 60 nanopillars) are used to validate the differentiation functionality. We can first obtain the $y$-component field intensity distributions after the Gaussian beam passes through the meta-differentiators by FDTD simulations, and then extract their corresponding angular spectrum distributions $|A_y|^2 / |A_y|^2_{\max}$ by performing the Fourier transform (Fig. 2d). It should be emphasized that the spot waist radius ($w = 3.5$ μm) is lower than the upper limit of 6 μm. The simulated results are perfectly consistent with the theoretical results, which suggests the image resolution defined in Supplementary Note 3 is rather conservative. $\hat{H} = C_1 \frac{\partial^2}{\partial k_x \partial k_y}$ results in zero intensity along $k_y$ and $k_x$ axes, and the maximum intensity is located on the two lines $\mathbf{k_x} + \mathbf{k_y}$ and $\mathbf{k_x} - \mathbf{k_y}$ (Fig. 2c, d). $\hat{H} = C_1 \frac{\partial}{\partial k_x}$ results in zero intensity along $k_y$ axis and the maximum intensity is located on $k_x$ axis only. $\hat{H} = C_1(\frac{\partial}{\partial k_x} + \frac{\partial}{\partial k_y}) = 2C_1 \frac{\partial}{\partial(k_x + k_y)}$ leads to zero intensity along the line $\mathbf{k_x} - \mathbf{k_y}$ and the maximum intensity appears on the line $\mathbf{k_x} + \mathbf{k_y}$. The theoretical and simulated results at discrete wavelengths within 450 nm and 1000 nm show similar field intensity distributions at 685.5 nm, suggesting the proposed meta-differentiators works with an extremely broad bandwidth (see Supplementary Note 4). Although the design method can be extended to arbitrary order angular spectrum meta-differentiators in principle, it poses

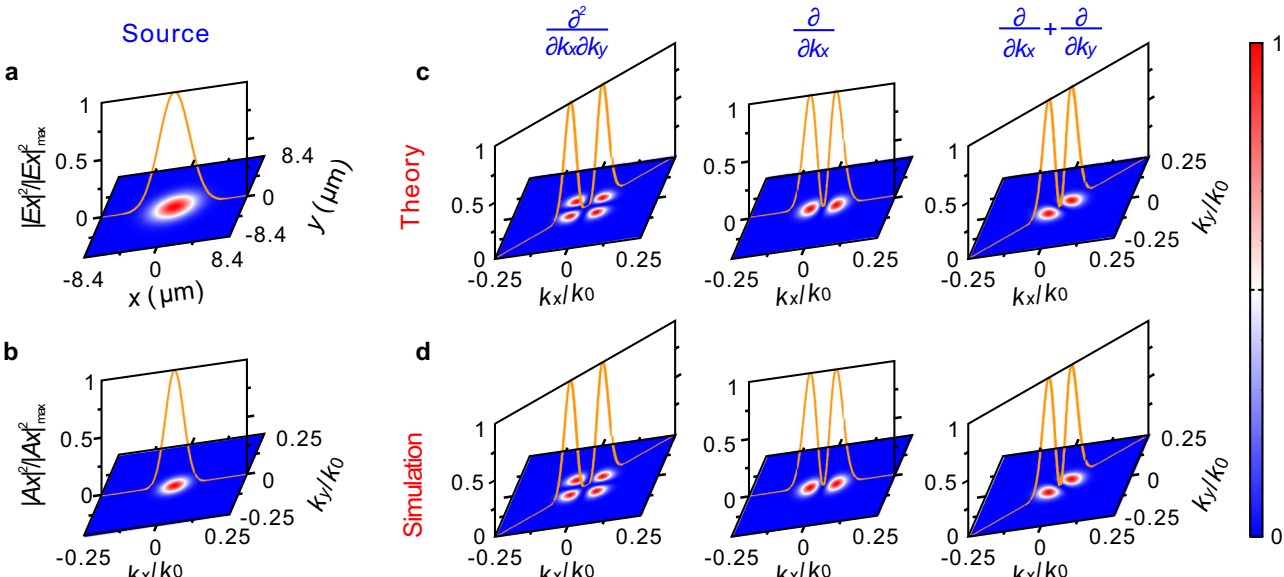

**Fig. 2 | Angular spectrum intensity distributions for three types of differentiations. a, b** Field intensity $|E_x|^2 / |E_x|^2_{\max}$ (**a**) and angular spectrum intensity $|A_x|^2 / |A_x|^2_{\max}$ (**b**) of the input $x$-polarized Gaussian beam with a waist radius of $w = 3.5$ μm at 685.5 nm. **c, d** Output angular spectrum intensity distributions for the three types of differentiations: theory (**c**) and simulation (**d**). $k_0$ is the wavenumber in the air.

bandwidth limitations for practically realizing higher order meta-differentiators. The field intensity offered by the meta-differentiators, especially for high order differentiation, is not always consistent with that predicted by theory, e. g., the output field is not symmetric in the long wavelength range, and the field intensity by the meta-differentiators is not so close to zero in the short wavelength range. The detailed study on bandwidth limitation for higher order angular spectrum meta-differentiators can be found in Supplementary Note 5.

## Experimental demonstration

We have fabricated three meta-differentiators of 300 μm × 300 μm in size (i.e., 1072 × 1072 nanopillars) on a 500 μm-thick silica substrate to verify the three types of angular spectrum differentiation. Figure 3a–c show scanning electron microscope (SEM) images taken at the lower right corner of each meta-differentiator. The experimental setup (Fig. 3d) is used to check if $t(x, y)$ exerted by the meta-differentiators is consistent with the theoretical values predicted by Eq. (2). The polarization directions of two polarizers are mutually orthogonal to extract

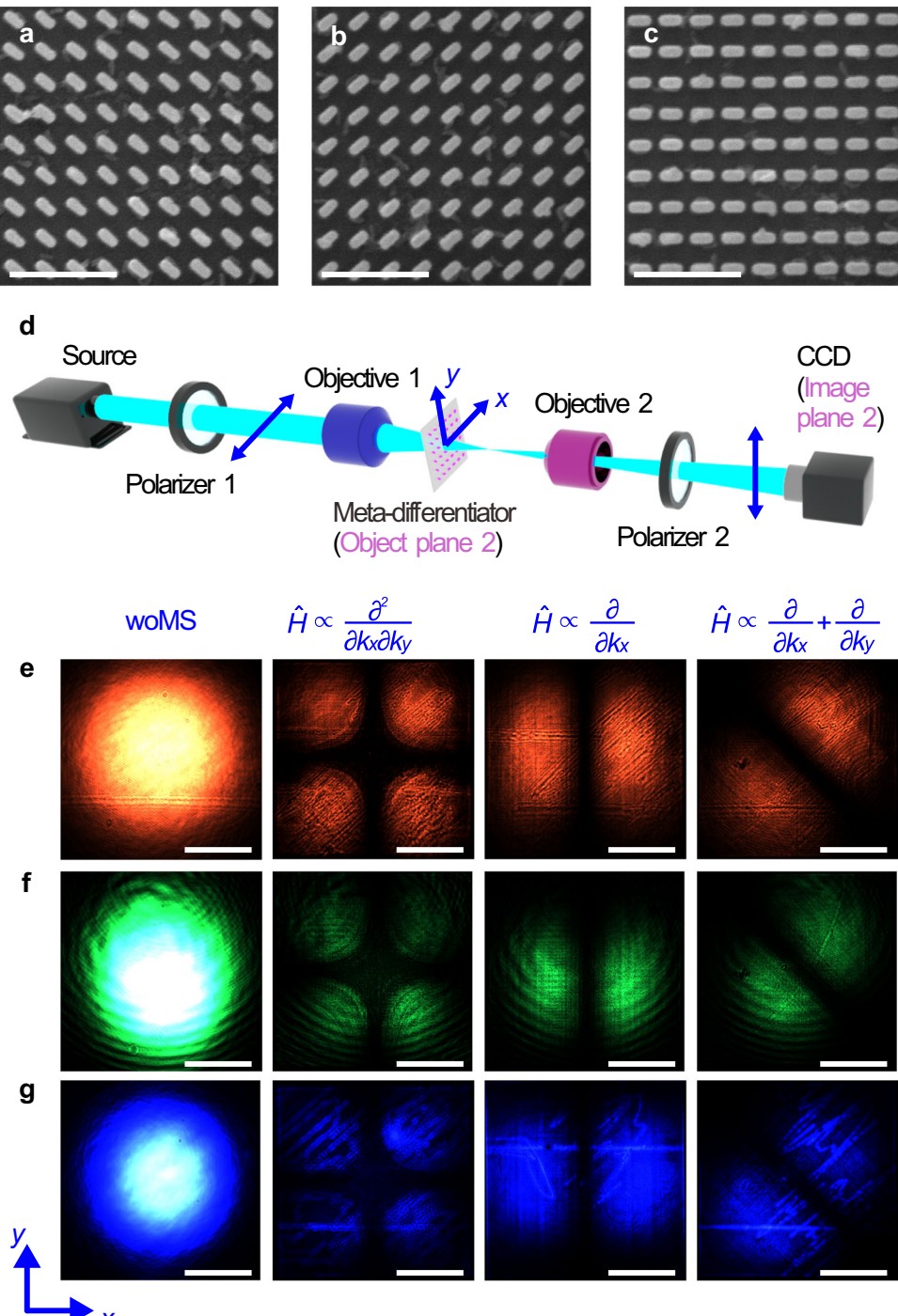

**Fig. 3 | Measured transmission field for three types of differentiations. a–c** SEM images of partial samples for angular spectrum differentiation with $\frac{\partial^2}{\partial k_x \partial k_y}$ (**a**) $\frac{\partial}{\partial k_x}$ (**b**) and $\frac{\partial}{\partial k_x} + \frac{\partial}{\partial k_y}$ (**c**). The scale bars are 1 μm. **d** Experimental setup for measurement. The meta-differentiator and the CCD camera are on the object and the image planes of objective 2, respectively. **e–g** Transmission field intensity distributions recorded by CCD with red light (685.5 nm) (**e**), green light (532 nm) (**f**) and blue light (450 nm) (**g**). The scale bars are 100 μm.

the cross-polarized transmission field. The distance between objective 1 and the meta-differentiator is adjusted to ensure that the light beam is slightly smaller than the meta-differentiator when reaching the left side of it. Under the illumination of red, green, and blue Gaussian light, the cross-polarized transmission field is recorded by the CCD camera (Fig. 3e–g). The output field intensity distributions are consistent with the theoretical results, as shown in Supplementary Note 6. The field intensity along lines $\mathbf{x}$ or $\mathbf{y}$, $\mathbf{y}$ and $\mathbf{x} - \mathbf{y}$, is zero, and is maximum along lines $\mathbf{x} + \mathbf{y}$ or $\mathbf{x} - \mathbf{y}$, $\mathbf{x}$ and $\mathbf{x} + \mathbf{y}$, corresponding to the three meta-differentiators with $\frac{\partial^2}{\partial k_x \partial k_y}$, $\frac{\partial}{\partial k_x}$, and $\frac{\partial}{\partial k_x} + \frac{\partial}{\partial k_y}$, respectively.

The meta-differentiator can implement differentiation operation on the angular spectrum of nontrivial objects. The test object is placed between the light source and the first polarizer (Fig. 4a). The distance between the object and objective 1 is first adjusted in order to image the object onto the meta-differentiator. The imaging area is slightly smaller than the meta-differentiator. Then objective 2 is moved to ensure the back focal plane of objective 1 is imaged onto the CCD camera via objective 2. In this scenario, the image recorded by the CCD camera corresponds to the differentiation of the object angular spectrum. The detailed explanation on how to use the experimental setup to retrieve the differentiation of the object angular spectrum can be found in Supplementary Note 7.

In our experiment, the object consists of a stainless steel plate drilled by three parallel rectangular holes and a single circular hole, which has a high transmission contrast between opaque and transparent regions. For the three parallel rectangular holes, the transmitted electric field can be written as

$$E_x(x,y) = \begin{cases} 1 & (|x|,|x \pm 2W_x| \le 0.5W_x, |y| \le 0.5W_y) \\ 0 & \text{otherwise} \end{cases} \quad (4)$$

The associated angular spectrum and differentiation with the three operations $\frac{\partial^2}{\partial k_x \partial k_y}$, $\frac{\partial}{\partial k_x}$ and $\frac{\partial}{\partial k_x} + \frac{\partial}{\partial k_y}$ can be found in Supplementary Note 8. The transmitted electric field after the object is plotted in the first panel of Fig. 4b. The associated normalized angular spectrum $|A_x(k_x, k_y)|^2$ is plotted in the first panel of Fig. 4c, with the intensity profile being shown in the first panel of Fig. 4d. The theoretical angular spectrum differentiations for $\frac{\partial^2}{\partial k_x \partial k_y}$, $\frac{\partial}{\partial k_x}$ and $\frac{\partial}{\partial k_x} + \frac{\partial}{\partial k_y}$ are shown in first panel of Fig. 4e, g, i, respectively, where the associated intensity profiles along lines A-B are plotted in the first panel of Fig. 4f, h, j, respectively. The angular spectrum at 685.5, 532 and 450 nm are experimentally extracted and recorded by the CCD camera, as shown in the second to the fourth panels in Fig. 4b-j. The experiment is well consistent with the above theoretical results. It can be seen that, under the angular spectrum differentiation of $\frac{\partial^2}{\partial k_x \partial k_y}$, a single row containing three main spots (Fig. 4c) is transformed into two rows, in which each row contains six main spots (Fig. 4e, f). Instead, when the operation $\frac{\partial}{\partial k_x}$ is considered, the three main spots in one row are only transformed into one row of six main spots (Fig. 4g, h). The angular spectrum experiences a first-order differentiation along $\mathbf{k_x} + \mathbf{k_y}$ direction with $\frac{\partial}{\partial k_x} + \frac{\partial}{\partial k_y}$, as $\frac{\partial}{\partial k_x} + \frac{\partial}{\partial k_y}$ is equivalent to $\frac{2\partial}{\partial(k_x + k_y)}$. As a result, under this operation each spot in one row is transformed into two spots along $\mathbf{k_x} + \mathbf{k_y}$ direction (Fig. 4i, j). It is worth discussing the antisymmetric profile for the angular spectrum intensity distributions at 450 nm. The polarizer has low extinction ratio of optical powers of perpendicular polarizations at shorter wavelengths, as opposed to longer wavelengths. The $y$-polarized light beam after polarizer 1 remains and reaches the CCD, which has an intensity comparable to the light beam contributing to angular spectrum differentiation. As the object is a single circular hole, the angular spectrum differentiation operation results are presented in Supplementary Note 9.

Derivative spectroscopy is a technique used to isolate weak spectral features from other unwanted strong features, and it can also be used to strengthen the contrast of a weak band to a strong overlapped signal in the spectral domain. It is widely used for trace analysis, purity test, and stability test[64,65]. While existing techniques operate in real space, our meta-differentiators can be used for isolating features in the angular spectrum domain of an object. If the object angular spectrum, $A_t(k_x, k_y)$, is much weaker than the additional background angular spectrum, $A_b(k_x, k_y)$, it is not possible to directly distinguish $A_t(k_x, k_y)$ and $A_b(k_x, k_y)$, i.e., one cannot detect the object angular spectrum for trace analysis. However, the object angular spectrum can be detected if the differentiation operation is applied on the object and background angular spectra, and the former differentiation value is much larger than the latter one. The corresponding experimental setup is established as shown in Fig. 5a, where one beam of light illuminates the object (path 1), and then is mixed with another beam of light (path 2), which mimics a strong background. In this case, $A_t(k_x, k_y)$ is associated with the angular spectrum of the three parallel rectangular holes, and $A_b(k_x, k_y)$ corresponds to the wide angular spectrum generated by a Gaussian-like source. $A_t(k_x, k_y)$ can be made much smaller than $A_b(k_x, k_y)$ by tuning a neutral density filter. In this situation, the mixed angular spectrum without the meta-differentiator around $k_x = 0$ is shown in Fig. 5b, d. It can be seen that the three spots are almost completely invisible (Fig. 5b, d). On the contrary, the six spots that define the angular spectrum decomposition of the target can be instead clearly observed around $k_x = 0$ after the mixed angular spectrum is processed by the meta-differentiator with $\frac{\partial}{\partial k_x}$ (Fig. 5c, e). This is because $A_t(k_x, k_y)$ features a maximum around $k_x = 0$, and the associated differentiation with respect to $k_x$ is close to zero, smaller than the differentiation of $A_b(k_x, k_y)$.

It should be emphasized that our angular spectrum analog meta-processor is also applicable to higher order differentiation, as has been shown in Supplementary Note 5. Moreover, while here we have focused on differential operations, the meta-processor may be extended to develop many other optical analog processing functionalities in the angular spectrum domain, such as differentiation, integration, convolution and equation solving. Spatial light modulators are also capable of providing spatially varying amplitude and phase distributions, and they are promising for achieving analogous angular spectrum operation functionalities. However, a pixel pitch of spatial light modulators is at least several wavelengths in size, and the resultant angular spectrum processors based on spatial light modulators may generate additional side lobes arising from high order diffraction. In addition to angular spectrum differentiation operation, the meta-differentiators may be explored to realize spatial differentiation operation, when combined with a 4 F system. Based on this functionality, we present high-contrast edge enhancement of resolution test charts and frog egg cells across the whole visible spectrum in Supplementary Note 10. By using nanopillars of varied cross-sections to provide full $0-2\pi$ phase coverage, it is possible to realize other types of angular spectrum differentiation operation, such as $\frac{\partial}{\partial k_x} + i\frac{\partial}{\partial k_y}$.

It should be emphasized that, while the mathematical operations demonstrated here operate on the complex amplitude of the angular spectrum of the electric field, the CCD camera used for our experiment can only record the field intensity information. Consequently, the phase information cannot be directly retrieved in our experiment. It is also worth discussing how the fabrication error affects the device performance. The silicon film has a surface roughness of 10 nm, and the fabrication uncertainty on the cross section of the silicon nanopillar can be controlled within 10 nm. Our analysis and simulation indicate that the fabrication error hardly influences the spatial resolution and the intensity of the output image. Finally, when assessing the performance of a passive computational metasurface, it is crucial to quantify the throughput efficiency, that is, how the intensity of the output image compares to the intensity of the input image. Different metrics can be

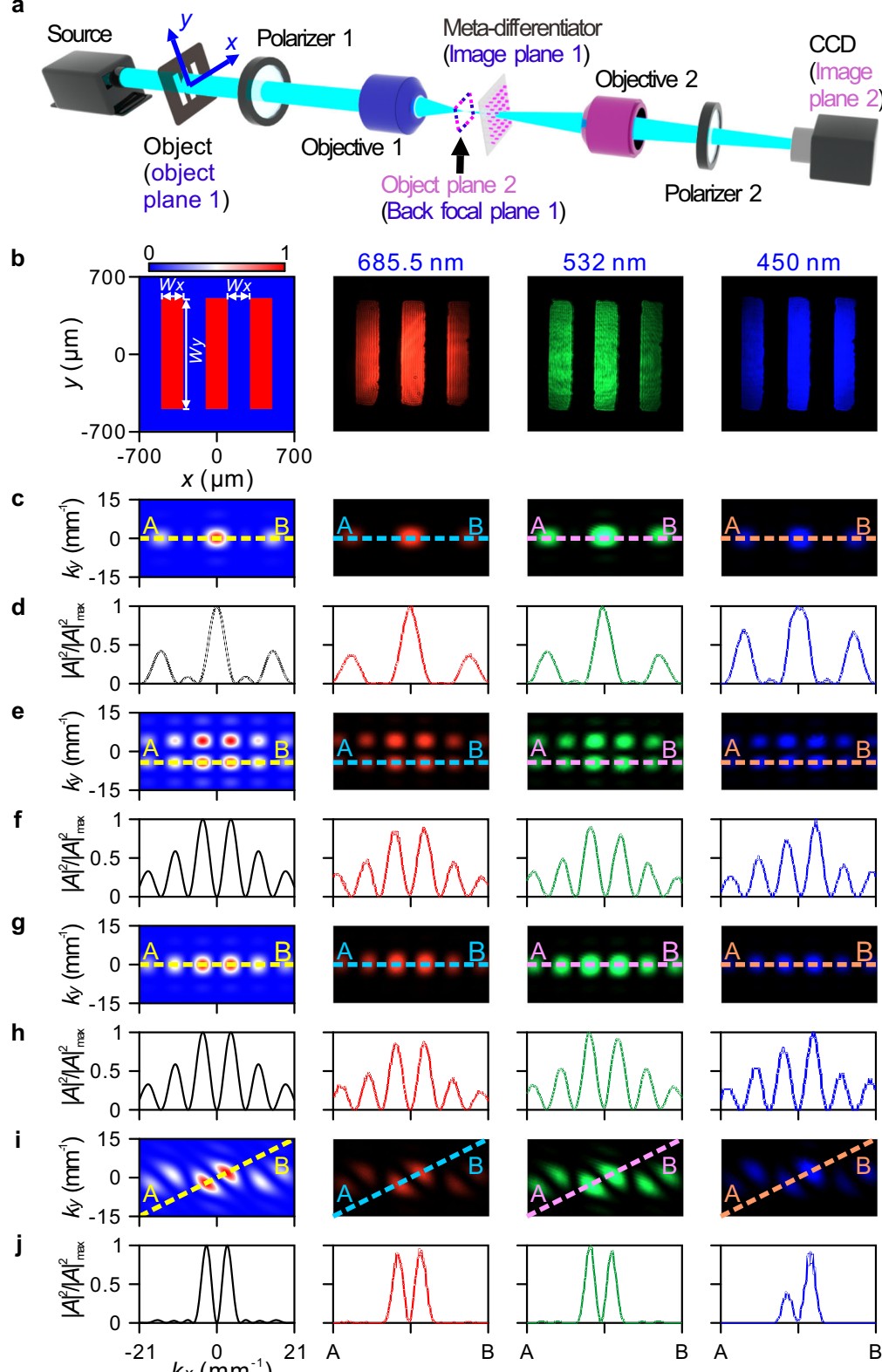

**Fig. 4 | Measured angular spectrum differentiations of three parallel rectangular holes. a** Experimental setup for performing angular spectrum operation of an object. The object and the meta-differentiator are on the object and the image planes of objective 1, respectively. The back focal plane of objective 1 is also the object plane of objective 2 and the CCD camera is on the image plane of the objective 2. **b** Theoretical (the first panel) and experimental (the second to fourth panels) intensity distributions of three parallel rectangular holes drilled in a stainless steel plate (1 mm thick). The geometrical parameters of the holes are set at $W_x = 200\ \mu m$ and $W_y = 1\ mm$. **c, d** Angular spectrum intensity distributions (**c**) and normalized intensity profiles along lines A-B in (**c**) (**d**). **e–j** Angular spectrum intensity distributions with $\frac{\partial^2}{\partial k_x \partial k_y}$ (**e**), $\frac{\partial}{\partial k_x}$ (**g**), and $\frac{\partial}{\partial k_x} + \frac{\partial}{\partial k_y}$ (**i**), and their normalized intensity profiles along lines A-B (**f**, **h**, **j**).

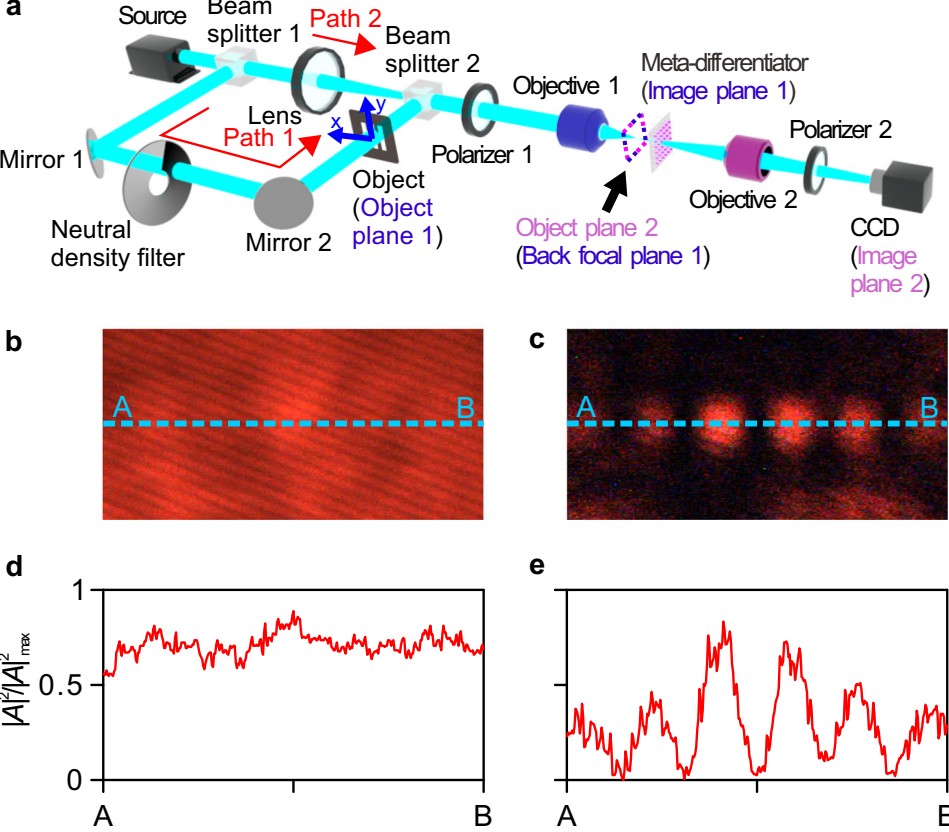

**Fig. 5 | Angular spectrum isolation experiment. a** Experimental setup for performing angular spectrum isolation at 685.5 nm. **b–e** Angular spectrum intensity distributions of the mixed light without (**b**) and with (**c**) the meta-differentiator with used to quantify the efficiency. The "peak efficiency" $\eta_{peak} = \max[I_{out}(x,$ $\frac{\partial}{\partial k_x}$. Their normalized angular spectrum intensity profiles along the lines A-B are shown in (**d**) and (**e**), respectively.

used to quantify the efficiency. The "peak efficiency" $\eta_{peak} = \max[I_{out}(x,$ $y)] / \max[I_{in}(x, y)]$, is defined as the ratio of the peak intensities in the output and input images, where $I_{out}(x, y)$ [$I_{in}(x, y)$] is the output (input) intensity map[45,46]. Another metric can be defined by considering the integrated intensities, i.e. $\eta_{int} = \iint_{-\infty}^{+\infty} I_{out}(x,y)\mathrm{d}x\mathrm{d}y / \iint_{-\infty}^{+\infty} I_{in}(x,y)\mathrm{d}x\mathrm{d}y$. We calculated both metrics for the measurements in Fig. 4. For both metrics, we obtained values in the order of $10^{-5}$ - $10^{-3}$, depending on the wavelength and the specific operation considered. The relatively low efficiencies are mainly attributed to the cross-polarized configuration of our setup, which leads to a rejection of a large fraction of the input power. The transmission efficiencies can be increased by using high-aspect-ratio dielectric metasurfaces made by low-loss materials, such as crystalline silicon and $TiO_2$. Detailed discussion regarding the transmission efficiency can be found in Supplementary Note 11.

## Discussion

In conclusion, we have demonstrated 'angular spectrum analog metaprocessors' that can perform differentiation of the angular spectrum of an object. The optical transfer function can be flexibly designed to target the desired angular spectrum processing functionalities. We have constructed three types of angular spectrum operations based on silicon metasurfaces, and demonstrated their capability of implementing angular spectrum differentiation on practical objects across the entire visible spectrum. We have also presented the use of a meta-differentiator to analyze target objects in the angular spectrum domain. Our results provide a novel approach to image processing in the angular spectrum domain, in contrast to previous strategies in the spatial domain. Such a design not only has broad application prospect for optical analog data processing and biological imaging, but also potentially enables new emerging optical detection techniques.

## Methods

### Fabrication

The sample fabrication process starts with the deposition of 220 nm-thick silicon film on a 500 μm silica substrate by using magnetron sputtering. A photoresist layer was spin-coated onto the silicon film and baked in an oven. Afterward, the desired patterns were defined on the photoresist layer by electron-beam lithography (EBL). After development of the photoresist, the pattern is transferred onto the silicon layer by inductively coupled plasma (ICP) etching. The targets with the three parallel rectangle holes and the single circular hole were drilled in the stainless steel plate by etching.

### Measurements

The setup to measure the metasurface transmission functions is shown in Fig. 3d. The light from a laser (red: DL-690-010-RS, green: MSL-FN-532-50 mW, blue: DL-450-010-RS) is collected by objective 1 (Olympus UPlanFL N, 10 × /0.30) and the distance between objective 1 and the meta-differentiators is adjusted to ensure that the light beam is slightly smaller than the meta-differentiators when reaching the left side of them. The light field on the meta-differentiators is photographed by objective 2 (Olympus UPlanFL N, 10 × /0.30) and the CCD camera (LBAS-U350-35C). The polarization directions of two polarizers are mutually orthogonal to extract the cross-polarized transmission field.

The setup to measure the angular spectrum differentiation is shown in Fig. 4a. The incident light passes through the object and the object is imaged on the meta-differentiator by objective 1. The object distance and the image distance are adjusted to ensure the image is smaller than the meta-differentiator. Then, the field on the back focal plane of the objective 1 is recorded by objective 2 and the CCD camera.

The polarization directions of two polarizers are mutually orthogonal to extract the cross-polarized transmission field.

The setup to measure the angular spectrum isolation is shown in Fig. 5a. The incident light is split into two beams (paths 1 and 2) by the beam splitter 1. The light along path 1 illuminates the object (i.e., the three parallel rectangle holes), and it is then imaged onto the meta-differentiator by objective 1. At the same time, the size of the spot from path 2 is adjusted by a lens (Thorlabs LB1471-A) and the spot is slightly larger than $W_x$. The intensity of the beam from path 1 is attenuated by the neutral density filter to make the corresponding angular spectrum almost completely covered by the wide angular spectrum from path 2. With the same method for measuring the angular spectrum mentioned in Fig. 4, the mixed angular spectrum without and with the meta-differentiators angular spectrum differentiation can be extracted.

## Data availability

All data required to interpret the results in this paper are provided within the main text and supplementary material. Any additional data in this study are available from the corresponding authors upon request.

## Code availability

The code that supports the plots within this paper is available from the corresponding authors upon request.

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

## Acknowledgements

L.C. is supported by National Key Research and Development Project of China (Grant No. 2021YFB2801903), National Natural Science Foundation of China (Grant No. 12074137), and Science, Technology and Innovation Commission of Shenzhen Municipality (Grant No. JCYJ20220530161010023). M.C., A.A. were supported by the Air Force Office of Scientific Research and the Simons Foundation. We thank Dr. Jun Su in the Optoelectronic Micro & Nano Fabrication and Characterizing Facility, WNLO, HUST for the SEM measurement.

## Author contributions

M.D., L.C. conceived the idea and initiated the work. L.C., A.A. guided the project. M.D. developed the theoretical framework, performed the numerical simulations and conducted the experiments. J.W., J.D., Z.R. discussed the results. M.D., L.C., M.C., A.A. wrote the manuscript and all authors reviewed the manuscript.

## Competing interests

The authors declare no competing interests.
