## [Peer Review File · Nature Communications]

Broadband angular spectrum differentiation using dielectric metasurfacesReviewer #1 (Remarks to the Author):

The authors propose to use dielectric metasurface based analog processors for three types of partial differentiation in angular spectra. The work is interesting. Detailed math derivations and metasurface designs are provided. The performance is extensively discussed. Here are a few comments for authors' references:

- The authors consider improving the rationale and clarifying the major achievements/key results and the novelty of the manuscript compared to the state of the art. The current statement is a bit too general (e.g. mention resolution, isolation effect etc.). The comparison to digital or the general comments of analogue processors can be left to introduction parts to detail.

- The authors have designed and experimentally demonstrated the spatial differentiation of three types, on a typical input of Gaussian wave with an optimized spot size, or with an amplitude mask. Is the results sensitive to the phase variations across the input pattern? It is unclear to the reviewer whether the mathematical operations are applied onto the electric field (complex number vary with phase) or the intensity/amplitude only. It is interesting to see the edge detection in bio samples (Fig. S11), which is mostly transparent. Does the result contain any phase information of the input image?

- How uniform is the film sputtering? Is geometric nonuniformity limiting the spatial resolution or attenuate the output image?

- Overall, the math/equations are more than most of the NC papers the reviewer read. The authors might consider to simplify the math and only leave critical/key parts in the main text. For example, on page 9, some intermediate math details and steps can be moved to SI

- What is the intensity of the transmitted image compared to the input? What is the percentage of the power of the cross-polarized output?

- It seems that the SEM images in Figure 3a-c correspond to Figure 1g-i. However, it's difficult to distinguish between the three types of Angular spectrum differentiations based on the device images. If the charging effect from the silicon oxide substrate is overwhelming, the authors may consider sputtering a thin film of metal (a few nm or less) to improve the visibility of the small features.

Minors:

- The abstract needs to be reworked – the comparison to digital counterparts read a bit strange 'Based on analog optical processing, this technique can provide an effective solution...'

-

Reviewer #2 (Remarks to the Author):

In this paper, the authors demonstrated the optical differentiation to angular spectrums of input light patterns. They did the theoretical calculation, numerical simulation and experiment. The results are consistent with each other. Compared with the existing wisdom, the current work made progress on the theme of optical analogue computation but regarding the importance, it cannot meet the standard for publication on a very top journal like Nature Communication at the current state. Previous works reported optical spatial differentiation mostly applying to object fields. The only difference here is that the operation was applied to the optical fields on the focal plane of a Fourier lens. Although the authors claim that they were first to impart mathematical operations on the angular spectrum, the novelty and application potential of this device are not clear yet. In addition, some questions may be considered if the authors transfer the paper to other journals.

1. Wenshuai Zhang et al [Zhang, W. et al. All-Optical Differentiator in Frequency Domain. Appl. Phys. Lett. 120, 011102 (2022)] have proposed a similar device to realize the first-order differentiation of

input signal in the temporal frequency domain. 2. The authors should compare their design with this previous works apart from the domain difference (spatial vs temporal).

The second sample has the same function with the third one, only by a 45-degree azimuthal angle rotation. It seems unnecessary to discuss the third one. In addition, is it possible to construct a two-dimensional operator with operations such as $\partial/(\partial k_x) + i \partial/(\partial k_y)$ and $\partial/(\partial k_r)$ using the metasurface adopted in this paper?

3. Optical spatial differentiation has been studied for a long time due to their essential application potential in edge detection. However, the actual application potential for the differentiation to the angular spectrum is not clear. The authors should elucidate this point in more details.

4. Compared with Figs. 5b and 5d, the higher contrast for plots in Figs. 5c and 5e seem straightforward as it performed the differentiation operation that could remove the strong dc signal coming from Path 2.

5. It is not clear that why the object plane of objective 2 should be coincident with the back focal plane of objective 1. Please address it.

Reviewer #3 (Remarks to the Author):

Image processing is of critical importance for various science and technology fields. This manuscript proposed dielectric metasurfaces operating across the whole visible spectrum which can be used as differentiators, realizing optical edge detection. The proposed design is also applicable to higher order differentiation and have potential applications for optical analog data processing and biological image.

Some comments are listed below:

1. In this work, 3 different metasurface differentiators are designed based on a same unit structure, a rectangular column. What should be considered when designing the unit structure, or in other words, the single cell?

2. In this work, the designed differentiator can take mixed partial derivatives of light field distribution with respect to x and y , or take partial derivatives separately. So it is not sensitive to gradients along some certain directions, like Fig. S9, Fig. S10, and Fig. S11. Why not consider using polar coordinates as a typical way, that means take the derivative with respect to r , where $r = (x^2 + y^2)^{1/2}$ is the radial component in polar coordinate, to get better results?

3. Figs. 3b and 3c are fuzzy, the detail in them is illegible, so it's hard to tell the difference between Figs. 3a-c. And the angle arrangement on photo Figs. 3a-c looks different from the schematic Figs. 1g-i. Can those pictures be updated?

4. In Figs. 4e-j, why the normalized intensity is not uniform after differentiation? For example, in Fig. 4j, the heights of the two peaks in the first chart (simulation) are the same, but the heights of the two peaks are different in the second chart (685.5nm) and the fourth chart (450nm). What is the reason for this phenomenon?

5. Fig. S3 and Fig. S4 both show the differential result with a wavelength of 1000 nm. Is it possible to supplement the relevant information in the 700-1000nm in Figs. S1c-d?

6. In this work, the intensity information presented is normalized. What is the actual intensity, or in other words, the efficiency of the designed differentiator?

7. Figure S8, what is the smallest linewidth that the differentiator can resolve? What is the resolution for edge detection?

8. In the supplementary material, it is mentioned that the bandwidth of higher-order differentiators is limited, but the explanation of this part about the limitation is not very clear, can it be described in detail? Is there any way to optimize it? Does it result from the performance of individual cells in different bands?

Response to reviewer # 1

The authors propose to use dielectric metasurface based analog processors for three types of partial differentiation in angular spectra. The work is interesting. Detailed math derivations and metasurface designs are provided. The performance is extensively discussed. Here are a few comments for authors' references:

We greatly appreciate the reviewer for the comments "The work is interesting. Detailed math derivations and metasurface designs are provided. The performance is extensively discussed". Below, we divide his/her comments into 7 topics, and reply each of them as follows:

1. The authors consider improving the rationale and clarifying the major achievements/key results and the novelty of the manuscript compared to the state of the art. The current statement is a bit too general (e.g. mention resolution, isolation effect etc.). The comparison to digital or the general comments of analogue processors can be left to introduction parts to detail.

Response: We are thankful to the reviewer for pointing out this issue. We have made detailed comparison with digital and other optical analogue processors in the introduction part, please see lines 39-44, 46-51, 61-69, and 70-73 in the main text.

2. The authors have designed and experimentally demonstrated the spatial differentiation of three types, on a typical input of Gaussian wave with an optimized spot size, or with an amplitude mask. Is the results sensitive to the phase variations across the input pattern? It is unclear to the reviewer whether the mathematical operations are applied onto the electric field (complex number vary with phase) or the intensity/amplitude only. It is interesting to see the edge detection in bio samples (Fig. S11), which is mostly transparent. Does the result contain any phase information of the input image?

Response: We appreciate the reviewer for raising this issue. Actually, the mathematical

operations are applied on the amplitude and phase of $A_p(k_x, k_y)$ of the incoming wavefront. We have improved the presentation to make this point clearer, please see lines 101-102 and 105 in the main text. Although the meta-differentiator can perform mathematical operations on complex fields in the Fourier domain, we cannot show the phase information in our experiment, since the CCD camera only records the field intensity information. To clarify this issue, we have added some discussion, please see lines 286-289 in the main text.

To confirm that the differentiation operation is applied on the complex amplitude, we have performed theoretical and numerical calculations to show that the phase variation can notably influence the output results. We have used a customized source with field pattern described by $E_x = \exp\left\{-\frac{x^2 + y^2}{(3.5\mu\text{m})^2} + 2i\pi \exp\left[-\frac{x^2 + y^2}{(3.5\mu\text{m})^2}\right]\right\}$. The field intensity distributions have the same amplitude as the one of Fig. 2 in the main text, but has a varied phase distribution, whereas the phase distribution used in Fig. 2 is uniform. The amplitude and phase distributions $|E_x|$ and $\angle E_x$ are shown in Figs. R1a, b, respectively. The associated amplitude and phase distributions $|A_x|/|A_x|_{\max}$ and $\angle A_x$ in the Fourier space are shown in Figs. R1c, d, respectively. The theoretical and simulated intensity distributions $|\hat{H}A_x|^2/|\hat{H}A_x|_{\max}^2$ for the three types of angular spectrum differentiation are shown in Figs. R1e, f. They are consistent, but quite different from those presented in Fig. 2c in the main text, which clearly indicates that the meta-differentiator performs the differentiation operation on the incident complex amplitude and the output results are sensitive to phase variations.

Fig. R1 | **a, b** Amplitude distribution of $|E_x|$ (**a**) and phase distribution of $\angle E_x$ (**b**) for

the field distribution $E_x = \exp\left\{-\frac{x^2 + y^2}{(3.5\mu\text{m})^2} + 2i\pi \exp\left[-\frac{x^2 + y^2}{(3.5\mu\text{m})^2}\right]\right\}$. **c, d** Associated

amplitude distributions $|A_x|/|A_x|_{\max}$ (**c**) and phase distributions $\angle A_x$ (**d**). **e, f** Output angular spectrum intensity distributions for three types of differentiations under customized incidence: theory (**e**) and simulation (**f**).

3. How uniform is the film sputtering? Is geometric nonuniformity limiting the spatial resolution or attenuate the output image?

Response: In this revision, we have measured the silicon film using a step profiler (DektakXT), which has a surface roughness of about 10 nm. The cross section of the fabricated silicon nanopillar is largely determined by the EBL, and the fabrication error can be controlled within 10 nm for the length and width of the nanopillar. We have also analyzed how the surface roughness and cross section of the silicon nanopillars affect the spatial resolution of the output image. Firstly, we find that the simulated output angular spectrum distributions with a surface roughness of 10 nm (see Fig. R2 below) are consistent with those presented in Fig. 2. As a result, the surface roughness of the silicon film hardly influences the spatial resolution and the output image. Secondly, we have studied how the cross section of the nanopillars affects the transmission coefficient, t_u and t_v , in Supplementary Fig. S1c. It can be seen that t_u and t_v change slightly with the cross section, if the variation of length and width of the nanopillar is less than 10 nm. Consequently, the cross-sectional nonuniformity slightly influences the intensity of the output image. In addition, C_1 does not affect the spatial resolution according to

Supplementary Eq. (S2) of Supplementary Note 3, although C_1 is proportional to $t_u - t_v$. As a result, the cross-sectional nonuniformity does not affect the spatial resolution. We have added a discussion on this issue in the revised manuscript, please see lines 289-293 in the main text.

Fig. R2 Simulated angular spectrum intensity distributions for the three types of differentiations with a surface roughness of 10 nm.

4. Overall, the math/equations are more than most of the NC papers the reviewer read. The authors might consider to simplify the math and only leave critical/key parts in the main text. For example, on page 9, some intermediate math details and steps can be moved to SI.

Response: We appreciate the reviewer for the valuable suggestions. In this revision, we have moved the derivation in page 10 into Supplementary Information, please see lines 231-232 in the main text and Supplementary Note 8.

5. What is the intensity of the transmitted image compared to the input? What is the percentage of the power of the cross-polarized output?

Response: We thank the reviewer for raising the issue of transmission efficiency. We note that Reviewer 3 also raised a similar comment. Indeed, when assessing the performance of a passive computational metasurface, it is crucial to quantify the throughput efficiency, that is, how the intensity of the output image compares to the intensity of the input image. To this aim, we considered two different quantitative metrics. First, we can use the so-called “peak efficiency”

$\eta_{peak} = \max[I_{out}(x, y)] / \max[I_{in}(x, y)]$, defined as the ratio of the peak intensities in the

output and input images, where $I_{out}(x, y)$ [$I_{in}(x, y)$] are the output (input) intensity maps^{A1,A2}. Additionally, we can consider a more “global” efficiency, defined as $\eta_{int} = \int \int_{-\infty}^{+\infty} I_{out}(x, y) dx dy / \int \int_{-\infty}^{+\infty} I_{in}(x, y) dx dy$. We have calculated both metrics for the measurements shown in Fig. 4. For both metrics, we obtained values in the order of $10^{-5} \sim 10^{-3}$, depending on the wavelength and the specific operation considered. These relatively low efficiencies are mainly attributed to the cross-polarized configuration of our setup, which leads to the rejection of a large fraction of the input power. The values are comparable to values obtained in other works that rely on polarization conversion^{A3-A5}. The transmission efficiencies can be increased by using high-aspect-ratio dielectric metasurfaces made by low-loss materials, such as crystalline silicon and TiO₂.

In this revision, we have added a new paragraph at page 12 to discuss these two different metrics and their values, and to discuss possible methods for enhancing the transmission efficiency, please see lines 293-304 in the main text and Supplementary Note 11.

- [A1] Cotrufo, M., Arora, A., Singh, S. & Alù, A. *Nat. commun.* **14**, 7078 (2023).
 [A2] Cotrufo, M., Singh, S., Arora, A., Majewski, A. & Alù, A. *Optica* **10**, 1331 (2023).
 [A3] Zhou, J. *et al. Natl. Sci. Rev.* **8**, nwaal76 (2021).
 [A4] Zhu, T. *et al. Phys. Rev. Appl.* **11**, 034043 (2019).
 [A5] Zhou, J. *et al. Proc. Natl. Acad. Sci. USA* **116**, 11137-11140 (2019).

6. It seems that the SEM images in Figure 3a-c correspond to Figure 1g-i. However, it's difficult to distinguish between the three types of Angular spectrum differentiations based on the device images. If the charging effect from the silicon oxide substrate is overwhelming, the authors may consider sputtering a thin film of metal (a few nm or less) to improve the visibility of the small features.

Response: We are thankful to the reviewer for raising this issue. We have sputtered the samples with a thin gold film and used SEM to validate the meta-differentiator

geometry. The SEM images were selected based on the lower right corners, at which the three types of angular spectrum meta-differentiators have the largest difference. In this revision, we have updated Figs. 3a-c and the associated discussion to clarify these issues, please see lines 208-209 and Fig. 3 in the main text.

7. Minors: - The abstract needs to be reworked – the comparison to digital counterparts read a bit strange ‘Based on analog optical processing, this technique can provide an effective solution.

Response: We fully agree with the reviewer on this point. This sentence has been changed to “Analog optical processing can provide an effective solution to perform large-scale and real-time data processing, superior to its digital counterparts, which have the disadvantages of low operation speed and large energy consumption”. Please see lines 23-24 in the revised abstract.

Response to reviewer #2

In this paper, the authors demonstrated the optical differentiation to angular spectrums of input light patterns. They did the theoretical calculation, numerical simulation and experiment. The results are consistent with each other. Compared with the existing wisdom, the current work made progress on the theme of optical analogue computation but regarding the importance, it cannot meet the standard for publication on a very top journal like Nature Communication at the current state. Previous works reported optical spatial differentiation mostly applying to object fields. The only difference here is that the operation was applied to the optical fields on the focal plane of a Fourier lens. Although the authors claim that they were first to impart mathematical operations on the angular spectrum, the novelty and application potential of this device are not clear yet. In addition, some questions may be considered if the authors transfer the paper to other journals.

We greatly appreciate the reviewer for the comments “Compared with the existing wisdom, the current work made progress on the theme of optical analogue computation but regarding the importance [...]. Although the authors claim that they were first to impart mathematical operations on the angular spectrum, the novelty and application potential of this device are not clear yet”. In this revision, we have significantly revised the manuscript to elaborate more on the importance, novelty and application potential of our work. Below, we divide his/her comments into 5 topics, and reply each of them as follows:

1. Wenshuai Zhang et al [Zhang, W. et al. All-Optical Differentiator in Frequency Domain. Appl. Phys. Lett. 120, 011102 (2022)] have proposed a similar device to realize the first-order differentiation of input signal in the temporal frequency domain.

Response: We are thankful to the reviewer for pointing out this paper related to analog signal processing, which helps us enriching the content of this submitted manuscript. We would like to point out that the scope of the paper mentioned by the reviewer is

very far from the scope of our paper. In that paper, a time-dependent signal is differentiated in time using the inverse Fourier transform of the frequency signal. In our submitted manuscript, we instead perform differentiation in the angular spectrum domain. Besides the different scopes, the practical implementation is also vastly different. The APL paper uses a conventional Soleil Babinet compensator based on birefringent crystals to complete the first-order differentiation of the input signal in the temporal frequency domain. In our submitted manuscript, we have used a compact method based on dielectric metasurfaces to realize differentiation in the angular spectrum domain. The angular spectrum analog meta-processor is also applicable to higher-order differentiation. In this revision, we have cited this reference and added a brief discussion to introduce this differentiator, please see Ref. 14 and lines 46-51 in the main text.

2. The authors should compare their design with this previous works apart from the domain difference (spatial vs temporal).

Response: We appreciate the reviewer for raising this issue. In this revision, we have made a comparison with previous works on optical analog processing. Specifically, we have explained how to realize digital signal processing with integrated electric circuits and why they suffer from low-speed and high-power consumption. Secondly, we have introduced papers working on optical analog processing in the spatial and temporal domain, which is realized with conventional 4F systems, interferometers and Soleil Babinet compensators, as well as with Bragg gratings and microring resonators. Thirdly, we have presented their performance and application potentials. In addition, we have made some presentations to emphasize the difference of our proposed meta-differentiator with previous spatial differentiators to highlight the novelty and application potentials. Please see lines 39-44, lines 46-51, lines 61-69, lines 70-73, and lines 89-94 in the main text.

The second sample has the same function with the third one, only by a 45-degree azimuthal angle rotation. It seems unnecessary to discuss the third one. In addition,

is it possible to construct a two-dimensional operator with operations such as $\partial/(\partial k_x) + i\partial/(\partial k_y)$ and $\partial/(\partial k_r)$ using the metasurface adopted in this paper?

Response: We are thankful to the reviewer for this important comment. It is true that, the differentiation function, $\frac{\partial}{\partial k_x} + \frac{\partial}{\partial k_y}$, can be obtained by rotating the meta-differentiator, corresponding to $\frac{\partial}{\partial k_x}$, by 45 degrees. Our purpose of designing and fabricating such a meta-differentiator with $\frac{\partial}{\partial k_x} + \frac{\partial}{\partial k_y}$ is to verify that our design principle is flexible to realize a sum over partial derivatives. **In this revision, we have emphasized this point, and shown how to realize a sum over higher-order partial derivatives by flexibly choosing l , C_j , and (m_j, n_j) , please see lines 127-132 in the main text.**

In principle, we can design an angular spectrum meta-differentiator to realize $\frac{\partial}{\partial k_x} + i\frac{\partial}{\partial k_y}$ by providing $t(x, y) = C_1(y - ix)$. The normalized amplitude and phase of $t(x, y)$ are shown in Figs. R3a, b, respectively. In contrast to the angular spectrum differentiation operation presented in our work, which can be realized with nanopillars having the same cross sections, realizing the angular spectrum differentiation operation $\frac{\partial}{\partial k_x} + i\frac{\partial}{\partial k_y}$ requires a full $0-2\pi$ phase coverage across the metasurface. As a result, nanopillars with varied cross-sections need to be used for establishing the meta-differentiator. Here, we have numerically demonstrated a meta-differentiator composed of 60×60 nanopillars for realizing $\frac{\partial}{\partial k_x} + i\frac{\partial}{\partial k_y}$. The theoretical and simulated results at 685.5 nm are shown in Figs. R3c-h, where the simulated results are consistent with the theoretical results.

Concerning the realization of $\partial/(\partial k_r)$, we find

$$\frac{\partial A_p(k_x, k_y)}{\partial k_r} = \frac{\partial A_p(k_x, k_y)}{\partial \sqrt{k_x^2 + k_y^2}} = \iint_{-\infty}^{+\infty} \frac{-i(k_x x + k_y y)}{\sqrt{k_x^2 + k_y^2}} E_p(x, y) \exp(-ik_x x - ik_y y) dx dy$$

in Cartesian coordinates. In this case, the metasurface needs to provide a real-space transfer function

$$t(x, y) = \frac{-i(k_x x + k_y y)}{\sqrt{k_x^2 + k_y^2}}$$

For any position (x, y) on the metasurface, $t(x, y)$ is not fixed as k_x

and k_y are variables. $t(x, y)$ is equivalent to $t(r, \varphi) = -ir \cos(\varphi - \phi)$ in polar coordinates, where $r = \sqrt{x^2 + y^2}$, $x = r \cos \varphi$, $y = r \sin \varphi$, $k_r = \sqrt{k_x^2 + k_y^2}$, $k_x = k_r \cos \phi$, $k_y = k_r \sin \phi$. $t(r, \varphi)$ is not only dependent on r and φ but also on ϕ . As a result, it may be more challenging to realize operations such as $\frac{\partial}{\partial k_r}$.

In this revision, we have added a discussion showing how to realize angular spectrum differentiation $\frac{\partial}{\partial k_x} + i \frac{\partial}{\partial k_y}$, please see lines 283-285 in the main text.

Fig. R3 a-b Amplitude (a) and phase (b) distributions of the real-space transfer function $t(x, y) = C_1(y - ix)$. **c-e** Amplitude $|A_y|$ (c), real part $\text{Re}[A_y]$ (d) and imaginary part $\text{Im}[A_y]$ (e) of the simulated output angular spectrum differentiation distributions with $\hat{H} = \frac{\partial}{\partial k_x} + i \frac{\partial}{\partial k_y}$. **f-h** Amplitude $|\hat{H}_x|$ (f), real part $\text{Re}[\hat{H}_x]$ (g) and imaginary part $\text{Im}[\hat{H}_x]$ (h) of the theoretical output angular spectrum differentiation distributions with $\hat{H} = \frac{\partial}{\partial k_x} + i \frac{\partial}{\partial k_y}$.

3. Optical spatial differentiation has been studied for a long time due to their essential application potential in edge detection. However, the actual application potential for the differentiation to the angular spectrum is not clear. The authors

should elucidate this point in more details.

Response: We are grateful to the reviewer for this important comment. It is true that optical spatial differentiation has been largely focused on the application potential in edge detection. Our manuscript has proposed to use angular spectrum meta-differentiators for enhancement of desired portions of an image within a mixed angular spectrum. In this revision, we have added more discussions to elucidate the application in more details, please see lines 256-271 in the main text.

4. Compared with Figs. 5b and 5d, the higher contrast for plots in Figs. 5c and 5e seem straightforward as it performed the differentiation operation that could remove the strong dc signal coming from Path 2.

Response: We are thankful to the reviewer for pointing out this issue. However, we are not sure if we have clearly understood his/her comment. We apologize if our original manuscript was confusing on the experimental details. As has been stated in our revision to your third comment, we have explained how the meta-differentiators process the angular spectrum differentiation on object and background angular spectra. The path 2 is associated with the background angular spectrum, which has Gaussian distributions along radial directions. The background angular spectrum has a maximum value around $k_x = 0$, leading to near zero differentiation around $k_x = 0$. Consequently, the angular spectrum intensity, associated with path 2, is removed in Figs.5c, e. We can only say that the background angular spectrum is suppressed near $k_x = 0$.

5. It is not clear that why the object plane of objective 2 should be coincident with the back focal plane of objective 1. Please address it.

Response: We thank the reviewer for this valuable suggestion. In our revision, we have established a theoretical model to explain why the image recorded by the CCD camera corresponds to the differentiation of the object angular spectrum, when the object plane of objective 2 is coincident with the back focal plane of objective 1. To clarify this issue, we have added a new Supplementary Note. Please see lines 223-226 in the main text, and Supplementary Note 7 in the revision.

Response to referee # 3

Image processing is of critical importance for various science and technology fields. This manuscript proposed dielectric metasurfaces operating across the whole visible spectrum which can be used as differentiators, realizing optical edge detection. The proposed design is also applicable to higher order differentiation and have potential applications for optical analog data processing and biological image.

We greatly appreciate the reviewer's comments "This manuscript proposed dielectric metasurfaces operating across the whole visible spectrum [...]. The proposed design is also applicable to higher order differentiation and have potential applications for optical analog data processing and biological image". Below, we divide his/her comments into 8 topics, and reply each of them as follows:

1. In this work, 3 different metasurface differentiators are designed based on a same unit structure, a rectangular column. What should be considered when designing the unit structure, or in other words, the single cell?

Response: We appreciate the reviewer for raising this issue. In the original manuscript, we have summarized the main points that should be considered when designing the unit structure, i.e., the silicon nanopillar:

- 1) We have obtained the real-space transfer function $t(x, y)$, associated with the angular spectrum differentiation operation \hat{H} . Please refer to lines 105-124 in the main text.
- 2) We have used the local transmission coefficient of the silicon nanopillar, $\sigma(x, y) = \frac{t_u - t_v}{2} \sin 2\theta(x, y)$, to satisfy the required $t(x, y)$. Please refer to the lines 137-160 in the main text and lines 61-67 in Supplementary Note 2.
- 3) To maximize the transmission efficiency at the three wavelengths (450, 532 and 685.5 nm), the cross-sectional geometry for the silicon nanopillar is optimized to obtain a large amplitude difference between t_u and t_v . Please refer to the lines 167-169 in the main text and the lines 75-83 in the Supplementary Note 2.

However, we have not explained why the cross section of the silicon nanopillar

is rectangular, rather than square, in the original manuscript. In this revision, we have clearly clarified this issue, please see lines 149-151 in the main text.

2. In this work, the designed differentiator can take mixed partial derivatives of light field distribution with respect to x and y, or take partial derivatives separately. So it is not sensitive to gradients along some certain directions, like Fig. S9, Fig. S10, and Fig. S11. Why not consider using polar coordinates as a typical way, that means take the derivative with respect to r, where $r = (x^2 + y^2)^{1/2}$ is the radial component in polar coordinate, to get better results?

Response: We appreciate the reviewer's valuable comment. Here, we explore the realization of the partial derivative using polar coordinates. Let us take

$$E_{out}(x, y) = \frac{\partial E_p(x, y)}{\partial \sqrt{x^2 + y^2}} = \frac{\partial E_p(x, y)}{\partial r} \quad \text{as an example, and derive the required real-space}$$

transfer function, $t(X, Y)$, that should be realized by the meta-differentiator. To simplify the mathematical derivation, it is assumed that $F_1 = F_2 = F$ in the 4F system (Supplementary Fig. S11). Based on the theoretical model in Supplementary Note 7, the output field can be expressed as

$$E_{out}(x, y) = -\frac{k_0^2 \exp(i4k_0F)}{4\pi^2 F^2} \iint_{-\infty}^{+\infty} t(X, Y) A_p\left(\frac{k_0X}{F}, \frac{k_0Y}{F}\right) \exp\left(-i\frac{k_0x}{F}X - i\frac{k_0y}{F}Y\right) dXdY \quad \text{in}$$

Cartesian coordinates. We can thus derive the required real-space transfer function as

$$t(X, Y) = -\exp(-i4k_0F) \frac{\iint_{-\infty}^{+\infty} \frac{\partial E_p(x, y)}{\partial \sqrt{x^2 + y^2}} \exp\left(i\frac{k_0X}{F}x + i\frac{k_0Y}{F}y\right) dx dy}{A_p\left(\frac{k_0X}{F}, \frac{k_0Y}{F}\right)}. \quad t(X, Y) \text{ can be simplified}$$

$$\text{as } t(X, Y) = \frac{k_0 \exp(-i4k_0F)}{2\pi F} \iint_{-\infty}^{+\infty} \frac{(Xx + x^2 + Yy + y^2) A_p\left(\frac{k_0x}{F}, \frac{k_0y}{F}\right)}{A_p\left(\frac{k_0X}{F}, \frac{k_0Y}{F}\right) [(X+x)^2 + (Y+y)^2]^{1.5}} dx dy. \quad t(X, Y) \text{ is}$$

equivalent to

$$t(R, \phi) = \frac{k_0 \exp(-i4k_0F)}{2\pi F} \int_0^{2\pi} \int_0^{+\infty} \frac{[r^3 + Rr^2 \cos(\phi - \phi)] A_p\left(\frac{k_0r}{F} \cos \phi, \frac{k_0r}{F} \cos \phi\right)}{A_p\left(\frac{k_0R}{F} \cos \phi, \frac{k_0R}{F} \cos \phi\right) [R^2 + r^2 + 2Rr \cos(\phi - \phi)]^{1.5}} dr d\phi \quad \text{in the}$$

polar coordinates, where $r = \sqrt{x^2 + y^2}$, $x = r \cos \phi$, $y = r \sin \phi$, $R = \sqrt{X^2 + Y^2}$, $X = R \cos \phi$, $Y = R \sin \phi$. $t(R, \phi)$ is not only dependent on R and ϕ but also r and ϕ , and hence the required $t(R, \phi)$ is varied with the incident light. As a result, it may be more challenging to construct the partial derivative $\frac{\partial}{\partial r}$, so as to realize edge-detection imaging, regardless of directions.

We have made some discussions to clarify this issue, please see lines 411-424 in the Supplementary Note 10.

3. Figs. 3b and 3c are fuzzy, the detail in them is illegible, so it's hard to tell the difference between Figs. 3a-c. And the angle arrangement on photo Figs. 3a-c looks different from the schematic Figs. 1g-i. Can those pictures be updated?

Response: We are thankful to the reviewer for raising this issue. We have sputtered the samples with a thin gold film and used SEM to observe the meta-differentiators. The SEM images are selected for the lower right corners, where the three types of angular spectrum meta-differentiators have the largest difference. In this revision, we have updated Figs. 3a-c and made some brief presentations, please see lines 208-209 and Fig. 3 in the main text.

4. In Figs. 4e-j, why the normalized intensity is not uniform after differentiation? For example, in Fig. 4j, the heights of the two peaks in the first chart (simulation) are the same, but the heights of the two peaks are different in the second chart (685.5nm) and the fourth chart (450nm). What is the reason for this phenomenon?

Response: We thank the reviewer for raising this issue. Our red light source (685.5 nm) is unstable, and bright fringes appear frequently when the output power is high. The unstable light source disturbs the differentiation images on the CCD. To avoid the asymmetrical distributions, we have used low output power to do the experiment, and have successfully obtained better field intensity distributions at 685.5 nm. Please see the second figure in Fig. 4j in the main text.

The asymmetrical field intensity distributions at 450 nm originate from polarizer 1,

which has low extinction ratio of optical powers of perpendicular polarizations at short wavelengths. The extinction ratio at long wavelengths (red and green light) is on the order of 10^5 , while it is on the order of 10^2 at blue light. We have used the power meter to record the x - and y -polarized light after polarizer 1. The ratio of the y -polarized light to the x -polarized light is on the order of 10^{-2} . After passing through the meta-differentiator, the x -polarized light (termed as the first light beam) is converted into y -polarized light, which is used for angular spectrum differentiation, with the transmission efficiency on the order of 10^{-5} . However, the y -polarized light (termed as the second light beam) after the meta-differentiator suffers from less loss, and its transmission efficiency is on the order of 10^{-3} . These two branches of light beams finally transmit through the polarizer 2 with no energy loss. As a result, they have comparable light intensity when reaching the CCD. The first light beam has an antisymmetric profile, while the second light beam exhibits a symmetric profile. The second light beam can significantly influence the angular spectrum differentiation of the first light beam, which results in the asymmetrical field intensity distributions. We have shown schematically below the output results when these two branches of light beams are combined (Fig. R4). In this revision, we have tried several different polarizers from different companies, and the field intensity distributions at 450 nm were not improved.

We have added some presentations to explain why the field intensity distributions at 450 nm have antisymmetric profile, please see lines 245-250 in the main text.

Fig. R4 Output results when the first (antisymmetric profile) and second (symmetric profile) light beams are combined.

5. Fig. S3 and Fig. S4 both show the differential result with a wavelength of 1000 nm. Is it possible to supplement the relevant information in the 700-1000nm in

Figs. S1c-d?

Response: We thank the reviewer for the valuable suggestion. We have supplemented the information in 700-1000 nm in Supplementary Figs. S1c, d in the revision.

6. In this work, the intensity information presented is normalized. What is the actual intensity, or in other words, the efficiency of the designed differentiator?

Response: We thank the reviewer for raising the issue of transmission efficiency. We note that Reviewer 1 also raised a similar comment.

Indeed, when assessing the performance of a passive computational metasurface, it is crucial to quantify the throughput efficiency, that is, how the intensity of the output image compares to the intensity of the input image. To this aim, we considered two different quantitative metrics. First, we can use the so-called “peak efficiency” $\eta_{peak} = \max[I_{out}(x, y)] / \max[I_{in}(x, y)]$, defined as the ratio of the peak intensities in the output and input images, where $I_{out}(x, y)$ [$I_{in}(x, y)$] are the output (input) intensity maps^{A1, A2}. Additionally, we can consider a more “global” efficiency, defined as $\eta_{int} = \int \int_{-\infty}^{+\infty} I_{out}(x, y) dx dy / \int \int_{-\infty}^{+\infty} I_{in}(x, y) dx dy$. We have calculated both metrics for the measurements shown in Fig. 4. The values of both metrics are of the order of 10^{-5} ~ 10^{-3} , depending on the wavelength and the specific operation considered. The relatively low efficiencies are mainly attributed to the cross-polarized configuration, which leads to a rejection of a large fraction of the input power. The values are comparable to values obtained in other works that rely on polarization conversion^{A3-A5}. The transmission efficiencies can be increased by using high-aspect-ratio dielectric metasurfaces made by low-loss materials, such as crystalline silicon and TiO₂.

In this revision, we have added a new paragraph at page 12 to discuss these two different metrics and their values, and to discuss possible methods for enhancing the transmission efficiency, please see lines 293-304 in the main text and Supplementary Note 11.

[A1] Cotrufo, M., Arora, A., Singh, S. & Alù, A. *Nat. commun.* **14**, 7078 (2023).

- [A2] Cotrufo, M., Singh, S., Arora, A., Majewski, A. & Alù, A. *Optica* **10**, 1331 (2023).
- [A3] Zhou, J. *et al. Natl. Sci. Rev.* **8**, nwaal76 (2021).
- [A4] Zhu, T. *et al. Phys. Rev. Appl.* **11**, 034043 (2019).
- [A5] Zhou, J. *et al. Proc. Natl. Acad. Sci. USA* **116**, 11137-11140 (2019).

7. Figure S8, what is the smallest linewidth that the differentiator can resolve? What is the resolution for edge detection?

Response: We thank the reviewer for pointing out this issue. We have used the negative resolution test chart to test the resolution in this revision. We have supplemented the resolution test results, please see Supplementary Fig. S12, Supplementary Table S4, and lines 433-441 in Supplementary Note 10.

8. In the supplementary material, it is mentioned that the bandwidth of higher-order differentiators is limited, but the explanation of this part about the limitation is not very clear, can it be described in detail? Is there any way to optimize it? Does it result from the performance of individual cells in different bands?

Response: We thank the reviewer for pointing out this issue. Actually, the bandwidth limitation does not result from the performance of individual cells in different bands. The reasons for the limited working bandwidth in the long and short wavelengths ranges are different. Below, we explain the reasons in detail and suggest methods to optimize the working bandwidth.

1. When designing the meta-differentiator, it is assumed that the coupling between the adjacent nanopillars is negligible. However, the coupling is enhanced when the meta-differentiator works at longer wavelengths, since the nanopillars have weaker localization on light, as opposed to short wavelengths. In this situation, $\sigma = \frac{t_u - t_v}{2} \sin 2\theta$ cannot precisely describe the transmission coefficients provide by the metasurfaces at long wavelengths. In other words, the transmission field provided by the metasurfaces

at long wavelengths is not consistent with the required values described by the real-space transfer function, $t(x, y) = \sum_{j=1}^l [C_j (-ix)^{m_j} (-iy)^{n_j}]$. Figure S5b clearly shows the deviation of field intensity distributions between the simulation and theory.

This deviation can be quantitatively evaluated by the overlap integral between the simulated and the theoretical transmitted fields^{A6},

$$\kappa = \frac{\left\{ \int \int_{-\infty}^{+\infty} [t(x, y) E_x] E_y^* dx dy \right\} \left\{ \int \int_{-\infty}^{+\infty} [t(x, y) E_x]^* E_y dx dy \right\}}{\left\{ \int \int_{-\infty}^{+\infty} [t(x, y) E_x] [t(x, y) E_x]^* dx dy \right\} \left\{ \int \int_{-\infty}^{+\infty} E_y E_y^* dx dy \right\}}. \text{ The calculated } \kappa \text{ for different}$$

differentiation orders at 1000 nm is presented in Supplementary Fig. S5b. The deviation becomes more apparent for higher order, as the orientation angle θ has more dramatic variation in the region far from the meta-differentiator center (see Fig. R5 below for $\left| \frac{d\theta}{dx} \right|$ as a function of x). Consequently, the designed higher order meta-differentiators have worse performance at long wavelengths.

Fig. R5 $\left| \frac{d\theta}{dx} \right|$ versus x for the three types of meta-differentiators.

Based on the lattice constant used in our paper, $P = 280$ nm, it might be challenging to achieve better differentiation performance at long wavelengths. An effective method of optimizing the differentiation performance at long wavelengths can be made by redesigning the meta-differentiators using a larger lattice constant. However, the resultant meta-differentiators have better performance at long wavelengths, but which is at the sacrifice of poorer performance at short wavelengths. In other words, it is

challenging to optimize the performance at long wavelengths, while keeping the central wavelength fixed.

2. As has been stated in the original Supplementary materials, the field intensity distributions offered by the metasurfaces around $x = 0$ is not consistent with theory (see Supplementary Fig. S6b in Supplementary Note 5). In our work, we have used the transmission coefficients for E_y under x -polarized incidence, $\sigma = \frac{t_u - t_v}{2} \sin 2\theta$, to design meta-differentiators, but the scattering from the vertices of the nanopillars is neglected. However, the E_y component, originating from the scattering around the vertices of the nanopillars, is dominant around $x = 0$ (see Supplementary Fig. S6c). Consequently, the simulated $|\sigma|$ deviates from the theoretical values when θ is around zero (see Supplementary Fig. S6d). This deviation will lead to the inconsistency of the differentiation field intensity distributions between the theory and simulation (see Supplementary Fig. S6a). For higher order meta-differentiators, there are more nanopillars, associated with small θ , arranged around $x = 0$ (see Fig. R6a below). As a result, the field intensity distributions offered by the metasurfaces around $x = 0$ show larger deviation for higher order differentiation, as opposed to lower order differentiation (see Supplementary Fig. S6b). This can be used to explain why the designed higher order meta-differentiators have worse performance at short wavelengths.

The feasible method of optimizing the differentiation performance at short wavelengths can be made by using low-loss crystalline silicon nanopillars to construct the meta-differentiators. In this case, we can optimize the crystalline silicon nanopillars to enhance $\left| \frac{t_u - t_v}{2} \right|$, so as to increase $|\sigma|$. As a result, the portion of E_y component originating from the scattering will be reduced. As has been stated in the response to your sixth comment, we have used crystalline silicon nanopillars to enhance the transmission efficiency of the meta-differentiators. The newly designed meta-differentiators with the crystalline silicon nanopillar with $W_u = 140$ nm, $W_v = 60$ nm, $H = 600$ nm and $P = 280$ nm, exhibit higher $\left| \frac{t_u - t_v}{2} \right|$ (see Supplementary Fig. S6e). We here present in Fig. R6b that, the simulated $|\sigma|$ coincides well with the theory. The

simulated angular spectrum differentiation distributions with the crystalline silicon at 450 nm are well consistent with the theory (see Figs. R6c, d). In contrast, the simulated angular spectrum differentiation distributions with the amorphous silicon (used for demonstrating angular spectrum differentiation in our original manuscript) show notable difference with the theory, especially for higher order differentiators.

In this revision, we have made some presentations to explain the reason for the bandwidth limitation of higher order differentiators, and discussed the feasible method of optimizing the bandwidth, please see Supplementary Figs. S6, S7, lines 188-202, and lines 219-242 in Supplementary Note 5.

Fig. R6 **a** θ versus x for the three types of meta-differentiators. **b** $|\sigma|/|\sigma|_{\max}$ versus θ as θ is around zero: theory (black dashed line), amorphous silicon nanopillar (yellow line) and crystalline silicon nanopillar (green line). **c** Distributions of the real part of the angular spectrum electric field at 450 nm. **d** Profiles of the real part of the angular spectrum electric field along $k_y = 0$ at 450 nm: theory (black dashed lines),

amorphous silicon nanopillar (yellow lines) and crystalline silicon nanopillar (green lines).

[A6] Zhou, N. *et al. Sci. Adv.* **5**, eaau9593 (2019).

Reviewer #1 (Remarks to the Author):

The authors' revisions fully address the reviewer's concerns.

Reviewer #2 (Remarks to the Author):

The manuscript has been improved in quality. However, the authors failed to answer the novelty of the current work. As the referee put before, analogy optical computation has been widely explored both theoretically and experimentally using dielectric metasurface in the literature. The core difference in this manuscript is that the authors moved the differentiating metasurface position from object plane to focal plane. Such a 4f-like system was initially considered in Solva's 2014 Science paper. But they have slight difference on metasurface design. From the discussions of the current manuscript, the referee did not see the special and important advantages to operate a field differentiation on Fourier plane, either.

Reviewer #3 (Remarks to the Author):

In the revised version of the manuscript, the authors have properly replied to all of questions raised during the first review process. This not only makes the manuscript more insightful, but also makes the whole frame easier to follow. Therefore, I think this manuscript can be accepted.

Response to reviewer # 1

The authors' revisions fully address the reviewer's concerns.

We are grateful to the reviewer's comments, and have made no further changes in this revision.

Response to reviewer #2

The manuscript has been improved in quality. However, the authors failed to answer the novelty of the current work. As the referee put before, analogy optical computation has been widely explored both theoretically and experimentally using dielectric metasurface in the literature. The core difference in this manuscript is that the authors moved the differentiating metasurface position from object plane to focal plane. Such a 4f-like system was initially considered in Solva's 2014 Science paper. But they have slight difference on metasurface design. From the discussions of the current manuscript, the referee did not see the special and important advantages to operate a field differentiation on Fourier plane, either.

We are thankful to the reviewer's comments. It is true that 4f systems have been extensively used for image processing, even without the aid of metasurfaces. However, we stress that, although our experimental setup might look like similar to a 4f system, we are actually performing differentiation in k-space. This is very different from the paper of Silva in 2014, which aims, like many others after that, to realize real-space differentiation on the incoming images (we also would like to emphasize that the Silva's 2014 Science paper did *not* consider a 4f system). We believe that the fundamental difference between real-space differential operations considered in previous work and the k-space differentiation introduced here is sufficiently clear in the current version of the manuscript. Therefore, we have made no further changes to this revision.

Response to referee # 3

In the revised version of the manuscript, the authors have properly replied to all of questions raised during the first review process. This not only makes the manuscript more insightful, but also makes the whole frame easier to follow. Therefore, I think this manuscript can be accepted.

We appreciate the reviewer's comments, and have made no further changes in this revision.